# COVID-19 Campus Closures in the United States: American Student Perceptions of Forced Transition to Remote Learning

Susan W. Parker, Mary A. Hansen and Carianne Bernadowski *

Department of Education, Faculty of the School of Nursing, Education, and Human Studies, Robert Morris University, Moon Township, PA 15668, USA; parkers@rmu.edu (S.W.P.); hansen@rmu.edu (M.A.H.)

* Correspondence: bernadowski@rmu.edu; Tel.: +1-412-397-5463

**Abstract:** As colleges and universities rapidly closed due to COVID-19, students and faculty were faced with unique challenges. The pandemic forced the cancellation of all campus activities, both extra-curricular and program-focused, such as student teaching experiences and nursing clinical rotations. Additionally, instructors were forced to rethink content delivery as coursework was quickly moved online and administered remotely via virtual platforms. Students were impacted as university level programs underwent a major paradigm shift within a matter of days or weeks. This study examined perspectives of undergraduate and graduate students regarding their experiences with rapid conversion from on-ground, in-person courses to remote instruction during the spring 2020 semester. The researchers employed a QUAN-QUAL descriptive mixed methods design. Using questionnaires and semi-structured interviews, the researchers examined general perspectives on in-person learning before the pandemic; initial perceptions about remote learning; and perceptions of the students about effort, engagement, needs, and ethical behavior as they engaged in totally remote learning. Results, analyzed using SPSS (QUAN) and inter-coder agreement (QUAL), indicated that initially students were engaged and satisfied with their in-person instruction, but became less satisfied and engaged during remote instruction. Undergraduate students experienced feelings of increased frustration, decreased accountability and engagement during remote learning, and turned to collaboration to earn points as they finished the semester.

**Keywords:** COVID-19; remote learning; higher education; student perspectives



## 1. Introduction

### 1.1. Thesis, Aims, Purpose of Article

The 2020 pandemic has presented day-to-day challenges across the globe (Neece et al. 2020) including many that modern society had never seen. In addition to concerns about personal health and financial stability, individuals had to be aware of community health and disease spread, quarantining, maintaining social distance, and the immediate closing of businesses, schools, and child-care centers. COVID-19 changed how the entire world operated in a matter of weeks to months. Everyone was affected in some way, and American colleges and universities were no exception. Keeping students safe was the top priority for institutions of higher education (IHE), but universities also had to pivot quickly to provide online and remote learning for the remainder of the spring 2020 semester. This move was made without precedent or research defining best practices to follow as higher education administration, faculty and students attempted to figure out how to organize, teach, learn, and maintain a campus atmosphere through a remote learning platform. Very little is known about the effects of COVID-19 on higher education (Chan 2020), although it is a growing topic of discussion (Fischer 2020). The paucity of research in this area supports the importance of current studies such as this one addressing this topic.

Compounding a difficult situation, many faculties had little to no experience teaching online. Research from the past few decades has often highlighted the difficulties of remote

learning. Olson and Olson (2000) determined that distance does matter. They identified four overarching conditions necessary for effective distance work: collaboration, technology, common ground and the nature of the work. Effective distance work is maximized only if all four conditions are met. Therefore, a great amount of common ground is necessary. The nature of the work should be clearly defined, but also loose, meaning participants have much autonomy. There should be a mutual readiness for collaboration in place, and participants need to be technology ready. A weak link in any of the four areas lessens the chance for effective distance work (Olson and Olson 2000).

Along with a focus on the importance of collaboration, a lack of connectedness is a concern for faculty and students in online and remote learning environments (Wells and Dellinger 2011). Students may choose online or remote course work for reasons such as convenience, work, child-care schedules, and less driving time to a campus location (Wells and Dellinger 2011). However, the more students perceive themselves as personally isolated, the less satisfied they generally are with web-based course work (Billings et al. 2001). Given that American university students were forced into isolation due to the quarantine rules brought on by the pandemic, and the students had previously chosen face to face, on ground courses, it makes sense that the majority of students were not happy with the quick move to remote and on-line learning.

Additionally, millennials have been described as students who view IHEs as a means to gaining employment rather than a training ground for critical and complex thinking (Buckner and Strawser 2016). This generational perspective tends to look externally for direction and approval, instead of regularly and actively taking responsibility for one's own learning (Buckner and Strawser 2016). Millennials are viewed to have less self-reliance than previous generations, and tend to be self-absorbed and egocentric with an overemphasis with earning a grade over learning the content (Buckner and Strawser 2016; Cain et al. 2012; Twenge 2009). Therefore, the purpose of this study was to examine American undergraduate and graduate student perspectives regarding their experiences with rapid conversion from on-ground, in-person courses to remote instruction during the spring 2020 semester. The specific research questions were: What were the students' general perceptions of learning at the start of the semester? What were students' perceptions of faculty preparedness, student effort, engagement, needs, and ethical behavior as they rapidly moved to remote instruction? Were there significant differences between the undergraduate and graduate students' perceptions?

The researchers theorized that students would perceive the transition as difficult as it was hypothesized that virtual learning would require sustained attention that could differ than in class coursework. However, the research team sought to learn what aspects of the transition were perceived as more difficult than others, and why. Because it is likely that remote and virtual learning will become a fabric of teaching and learning in the future, gathering and analyzing data could assist faculty at IHE in devising online and virtual coursework to best support students in future course offerings.

The COVID-19 pandemic forced higher education institutions to rethink traditional pedagogy and modalities relative to teaching content. Adapting quickly, university classrooms capitalized on pedagogies such as flipped classrooms (Nerantzi 2020; Zheng and Zhang 2020), peer learning (Mohammed et al. 2020), and peer teaching (Jeong et al. 2020). Research on student performance, motivation (Aguilera-Hermida 2020), and student perceptions can also be found in recent research. A review of the literature also reflects a wealth of literature available in the fields of international medicine and dentistry (Savage et al. 2020; Srinivasan 2020; Torda 2020), but very little relative to higher education or students' perspectives. The current literature available centered on technology, faculty perceptions, and student perceptions (including student mental health).

### 1.1.1. Technology

Access to laptops, consistent and reliable Wi-Fi, and internet access is examined (Schaffhauser 2020) in the literature. The pandemic has placed a spotlight on the educa-

tional divide, as COVID-19 has had a devastating effect on students who live in communities that are rural and poor (Berry 2020), making educational and socioeconomic disparities abundantly clear. Additionally, specific technology platforms have been explored, such as Microsoft Teams, Canvas, Zoom, and Google Hangouts (Nic Dhonncha and Murphy 2020; Wyse et al. 2020). Faculty and students have weighed in on discussions of strengths and weaknesses of different platforms including which are easiest to navigate and user friendly. Learning through social media such as Facebook and Twitter, online videos and resources, augmented reality (AR) and virtual reality (VR) have been used as teaching tools before the pandemic. However, the pandemic has made teaching and learning through virtual tools a necessity and not a choice.

Although there are positive attributes to remote learning (for example, access to lecturers who cannot physically visit a campus), technology renders almost every test "open book" (Nic Dhonncha and Murphy 2020). With students having access to information at their literal fingertips, accurate measurement of student performance becomes challenging. Easy to implement suggestions have included strictly limiting time allowed to complete exams, and using images or other materials that may be difficult to search for online (Nic Dhonncha and Murphy 2020). Larger universities may have the option of utilizing third party exam proctoring systems such as ProctorU or Examinty as suggested in Iyer et al. (2020), but smaller schools often do not have the resources to implement learning management systems for exam monitoring and beyond. In Kuwait, private universities transitioned to remote learning, but public universities postponed spring semesters until August 2020 (Al-Taweel et al. 2020). The decision to postpone was multifaceted, and included regulations surrounding accreditation of online programs, and the perception that online education is inferior to face-to-face course work. In general, there are still unanswered questions regarding how to best utilize technology for learning.

### 1.1.2. Faculty Perceptions

Due to the speed with which the virus spread, the almost instantaneous shift from face-to-face teaching to remote coursework did not allow faculty to prepare adequately (Al-Taweel et al. 2020). Moreover, faculty have had generally negative attitudes about remote instruction, specifically in courses where "hands-on" experiences, such as fieldwork and labs, are utilized (Barton 2020; Nic Dhonncha and Murphy 2020), although some remote teaching methods were viewed as effective (Barton 2020) such as asynchronous instruction that involves little to no interaction with the instructor or peer.

In a study out of Singapore, lecturers teaching anatomy felt positively about the use of Zoom and narrated Powerpoint presentations to teach medical students about the pharynx, larynx and ear (Srinivasan 2020). Anatomy students have expressed interest in using virtual reality (VR) platforms (Saverino 2020; Triepels et al. 2020), concluding that they can visualize anatomical structures three dimensionally, providing the ability to view organs from different perspectives (Iwanaga et al. 2020). In Ireland, teaching medical students about dermatology was facilitated online using video lectures, virtual collaboration tools, real-time discussions, document sharing, and student led case study presentations (Nic Dhonncha and Murphy 2020). Many faculties feel that remote learning has many positives, but cannot replace face-to-face teaching. However, remote teaching and learning are here maybe for the long term, and in some ways have prompted innovations in teaching methods (Nic Dhonncha and Murphy 2020; Walsh et al. 2020).

### 1.1.3. Student Perceptions

Students have discussed the convenience of learning remotely (Branquinho et al. 2020). However, in one study of medical students at Duke University, over two thirds of study participants (116/179 students) expressed a desire to return to face-to-face learning (Compton et al. 2020). There has been an examination of standardized testing and the lack of access to testing and testing sites (Gewertz 2020; Lambert 2020). Lack of access may have an effect on student graduation, certification and admittance to higher education and

even graduate school. Students have also lamented the loss of rites of passage such as graduation ceremonies (Branquinho et al. 2020). Moreover, they have shared the impact the pandemic has had on their social lives, lack of daily routines including sleep and exercise, additional screen time, and stress, and how these changes have affected their productivity (Abbas et al. 2020; Branquinho et al. 2020).

Student mental health is another growing concern, with one study finding 71% of university students interviewed indicated stress and anxiety due to the COVID-19 outbreak (Son et al. 2020). Identified stressors included fear about one's health, or the health of a loved one, decreased social interactions, and worry about academic performance. In the early spring of 2020, it was estimated that one person in five worldwide was living under some form of lockdown conditions (Davidson 2020) leading scientists to describe the social conditions surrounding the pandemic as the largest psychological experiment ever to be conducted (Van Hoof 2020).

Students have expressed some positive lessons to note during the pandemic such as realizing the importance and strengthening of relationships and having more leisure time with school closures (Branquinho et al. 2020). With so many unknowns regarding the pandemic, and current thought leaning toward the fear that achievement gaps will continue to exist (Wyse et al. 2020), high schools, colleges and universities in the United States and beyond are struggling to determine the most appropriate and effective course of action for all stakeholders. Therefore, it is prudent to continue to gather data from all affected parties, including students.

## 2. Materials and Methods

The researchers implemented a QUAN-QUAL descriptive mixed methods study involving cross-sectional survey research and semi-structured, in-depth interviews. The target population for the survey portion of the study consisted of undergraduate and graduate students in the United States who were enrolled in on-ground courses during the Spring 2020 semester and were forced to abruptly transition to remote learning amid the COVID-19 pandemic. The qualitative portion of the study focused on undergraduate students from this same population. The researchers focused only on undergraduate students for the qualitative portion of this study to gain a more in-depth perspective on the impact of their transition to remote learning. Preliminary survey results suggested that undergraduate students were less engaged in remote learning and cheated more than graduate students, so the researchers intentionally decided to focus interviews on undergraduate students to provide more in-depth information about these topics.

### 2.1. Participants

Potential survey respondents were initially contacted through university course roster student email addresses. Initial contacts were asked to "tell your friends to complete the survey" which was also posted on social media forums. Snowball sampling was chosen due to the pandemic. While there is no guarantee that using social media platforms and snowball sampling ensures all the respondents were university students, there is also little reason to believe individuals not pursuing a university degree would be interested in completing the survey.

A total of 80 individuals responded to the questionnaire. The majority of respondents were female ($n = 62$, 77.5%), while 15 (18.8%) were male and 2 (2.5%) preferred not to provide their gender. Slightly more respondents attended private schools ($n = 45$, 56.3%) versus public ($n = 34$, 42.5%), but a Chi-square goodness of fit test showed the proportions of respondents from these two types of institutions were not significantly different, $\chi^2(1) = 1.532$, $p = 0.216$. Respondents were from institutions of various sizes, with 25 (31.3%) from schools with undergraduate enrollment less than 5000 (small); 34 (42.5%) from schools with enrollments ranging from 5000 to 20,000 (medium); and 20 (25.0%) from schools with 20,000 (large) or more undergraduates. Both undergraduate ($n = 57$, 71.3%) and graduate ($n = 22$, 27.5%) students were represented in the study, with 11 freshmen

(13.8%), 15 sophomores (18.8%), 23 juniors (28.7%), eight seniors (10.0%), five master's level students (6.3%), and 17 doctoral students (21.3%). Overall, respondents at a variety of educational levels, with different genders, and from different sizes of institutions were represented in the study.

Ten college students, who attend both small and large private institutions and two large public institutions, were interviewed. Participants were recruited via purposeful sampling. All interviewees also participated in the survey portion of the study. Class standing of participants was varied and included sophomores, juniors and seniors. Table 1 illustrates the demographic characteristics of participants.

**Table 1.** Participant Demographics.

| Participant | Gender | Year | Status | Type | Size |
|---|---|---|---|---|---|
| 1 | F | Rising senior | Commuter | Private | Small |
| 2 | F | Rising senior | Resident | Private | Small |
| 3 | F | Rising senior | Commuter | Private | Small |
| 4 | F | Rising senior | Resident | Private | Small |
| 5 | F | Rising senior | Resident | Private | Small |
| 6 | F | Rising senior | Resident | Private | Small |
| 7 | F | Rising junior | Resident | Public | Large |
| 8 | F | Rising junior | Resident | Public | Large |
| 9 | F | Rising junior | Resident | Private | Small |
| 10 | F | Rising sophomore | Resident | Private | Large |

*2.2. Instruments*

Self-constructed instruments were used to collect survey and interview data. The questionnaire was designed to measure several dimensions. Five-point Likert scales with response options strongly disagree (1), disagree (2), neutral (3), agree (4) and strongly agree (5) were used for all closed-response items. Items where agreement meant a negative behavior (e.g., I cheated) were reverse coded for the reliability analysis. The scales showed high internal consistency reliability overall (46 items, $\alpha = 0.895$) and by subscale: general perspectives on in-person learning before the pandemic (10 items, $\alpha = 0.815$); general perspectives on remote learning before the pandemic (10 items, $\alpha = 0.830$); initial perceptions about remote learning (8 items, $\alpha = 0.659$); and perceptions of the students about effort and engagement (8 items, $\alpha = 0.818$); and ethical behavior (5 items, $\alpha = 0.906$). The five additional Likert items that were asked were not related to a single subscale, but they were included in the reliability computation of the overall instrument.

Additionally, three open-ended items were administered to understand students' general perceptions of remote learning, what students liked about their transition to remote leaning, and suggestions they had for improving future remote learning experiences. The semi-structured interview protocol consisted of ten open-ended questions that were crafted to examine participants' impressions of the transition to virtual learning, comfort levels, levels of engagement, and advantages and disadvantages of the modality. Furthermore, one question examined the factors that aided or hindered the students' course work including teaching styles and overall satisfaction with the instructor.

A researcher-created semi-structured interview protocol was developed based on the questionnaire with an emphasis on students' perceptions of the transition from on-ground, in-person learning to virtual, the obstacles faced with this transition, and participants' levels of engagement.

*2.3. Procedures*

The questionnaire was administered using an online survey package. The link was shared via several social media platforms for four weeks from May to June 2020. A total of 449 individuals viewed the questionnaire; 124 individuals started the questionnaire, and 80 completed the questionnaire. Descriptive and inferential statistics were computed using

SPSS, and open-ended responses were hand-coded. Semi-structured interviews were conducted virtually through Google Hangout™ and recorded with permission. Once transcribed, all recordings were deleted, and data analysis occurred by utilizing intercoder-agreement (Tinsley and Weiss 2000), which improved the systematic analysis and transparency of the coding process. By utilizing intercoder-agreement, the researchers were able to improve the trustworthiness of the data. This process included each researcher coding the data separately, sharing their analysis, discussing their findings and developing a consensus on the results.

## 3. Results

The purpose of the current study was to understand undergraduate and graduate students' experiences with rapid conversion from on-ground, in-person courses to remote instruction during the Spring 2020 semester. Using mixed methodology, the researchers examined attitudes and behaviors of students overall and by several demographic variables including level (undergraduate versus graduate), type of institution (public versus private), student status (commuter versus non-commuter), and size of institution (less than 5000, 5000-less to 20,000, more than 20,000). The researchers examined general perspectives on in-person learning before the pandemic; initial perceptions about remote learning; and perceptions of the students about effort, engagement, needs, and ethical behavior.

Results from the questionnaire data for all respondents are presented first, followed by results based on demographic subgroups. Results from the interviews follow, including overlapping and unique findings from the two data sources.

### 3.1. Findings: Questionnaire Responses

Tables 2–7 show results from all respondents to the Likert-response questionnaire items. For all items, percentages for disagree and strongly disagree options, as well as the agree/strongly agree items, were combined for presentation, while other descriptive statistics reflect the five point scales. Counts, percentages, medians, means, standard deviations, and sample sizes of respondents are provided for each questionnaire item.

To provide context about student perceptions related to instruction before their abrupt switch to remote learning, respondents were asked for their views about their own and their professors' behaviors. Table 1 shows that for all items except two, 90% or more of respondents agreed or strongly agreed that they attended classes and completed coursework; were challenged and engaged; had professors that used effective instructional and assessment strategies and were engaged. Overall, over 92.4% ($n = 73$) of students agreed or strongly agreed that they were satisfied with their in-person instruction before the switch to remote learning occurred. Additionally, the vast majority of respondents disagreed or strongly disagreed ($n = 69$, 87.3%) that they cheated during in-person instruction. Finally, the lowest percentage of agreement, still representing the vast majority, reflected by the 83.5% ($n = 66$) of respondents who agreed or strongly agreed that their in-person instruction was worth the amount they paid. Overall, respondents agreed that they and their professors were engaged with their instruction.

Table 3 presents a parallel set of questions about remote learning. Overall, results showed that the vast majority of students agreed that they attended their live, remote classes ($n = 70$, 88.6%) and completed the majority of their classwork ($n = 78$, 97.5%). However, students' levels of agreement with other critical aspects of remote learning was not as high, and marked a change from their responses about in-person learning. For instance, only approximately half of the respondents agreed or strongly agreed that they were challenged by their remote course work ($n = 38$, 47.5%) and that their professors were engaged for their remote course work ($n = 42$, 54.5%). Closer to 40% agreed that their professors used instructional and assessment strategies effectively. Further, approximately half of the respondents disagreed or strongly disagreed that they were engaged in their remote courses and were satisfied with their remote course work.

Telling is the fact that a strong majority of respondents ($n = 54$, 69.2%) disagreed that they were satisfied with their remote course work. Of further concern is that more than

one-third of students agreed or strongly agreed that they cheated during online instruction (*n* = 29, 36.7%) while half disagreed. With a tenth of the students responding with a neutral response for this item, the researchers suspect the occurrence of cheating could have been even higher.

**Table 2.** General Perceptions of Learning: In-Person.

| Item. | Strongly Disagree/Disagree | | Neutral | | Agree/Strongly Agree | | Summary Statistics | | | |
|---|---|---|---|---|---|---|---|---|---|---|
| | *n* | % | *n* | % | *N* | % | *Mdn* | *M* | *SD* | *n* |
| I attended the vast majority of in-person classes. | 2 | 2.5% | 1 | 1.3% | 77 | 96.3% | 3 | 2.9 | 0.33 | 80 |
| I was challenged by my in-person course work. | | | 8 | 10.0% | 72 | 90.0% | 3 | 2.9 | 0.30 | 80 |
| I completed the vast majority of my in-person course work. | | | 2 | 2.5% | 77 | 97.5% | 3 | 3.0 | 0.16 | 79 |
| I was engaged in my in-person courses. | | | 5 | 6.4% | 73 | 93.6% | 3 | 2.9 | 0.25 | 78 |
| I cheated during my in-person courses. | 69 | 87.3% | 6 | 7.6% | 4 | 5.1% | 1 | 1.2 | 0.50 | 79 |
| My professors used effective instructional strategies during my in-person course work. | | | 7 | 8.9% | 72 | 91.1% | 3 | 2.9 | 0.29 | 79 |
| My professors used effective assessment strategies during my in-person course work. | 2 | 2.6% | 5 | 6.4% | 71 | 91.0% | 3 | 2.9 | 0.39 | 78 |
| My professors were engaged for my in-person course work. | | | 3 | 3.8% | 76 | 96.2% | 3 | 3.0 | 0.19 | 79 |
| My in-person instruction was worth the amount I paid. | 7 | 8.9% | 6 | 7.6% | 66 | 83.5% | 3 | 2.7 | 0.61 | 79 |
| I was satisfied with my in-person course work. | 1 | 1.3% | 5 | 6.3% | 73 | 92.4% | 3 | 2.9 | 0.33 | 79 |

*Mdn*: Median. *M*: Mean. *SD*: Standard Deviation. *n*: sample size.

**Table 3.** General Perceptions of Learning: Remote.

| Item. | Strongly Disagree/Disagree | | Neutral | | Agree/Strongly Agree | | Summary Statistics | | | |
|---|---|---|---|---|---|---|---|---|---|---|
| | *n* | % | *n* | % | *n* | % | *Mdn* | *M* | *SD* | *n* |
| I attended the vast majority of live, remote classes. | 7 | 8.9% | 2 | 2.5% | 70 | 88.6% | 3 | 2.8 | 0.59 | 79 |
| I was challenged by my remote course work. | 26 | 32.5% | 16 | 20.0% | 38 | 47.5% | 2 | 2.2 | 0.89 | 80 |
| I completed the vast majority of my remote course work. | 1 | 1.3% | 1 | 1.3% | 78 | 97.5% | 3 | 3.0 | 0.25 | 80 |
| I was engaged in my remote courses. | 37 | 46.3% | 20 | 25.0% | 23 | 28.8% | 2 | 1.8 | 0.85 | 80 |
| I cheated during my remote courses. | 41 | 51.9% | 9 | 11.4% | 29 | 36.7% | 1 | 1.8 | 0.93 | 79 |
| My professors used effective instructional strategies during my remote course work. | 29 | 36.3% | 19 | 23.8% | 32 | 40.0% | 2 | 2.0 | 0.88 | 80 |
| My professors used effective assessment strategies during my remote course work. | 25 | 31.3% | 21 | 26.3% | 34 | 42.5% | 2 | 2.1 | 0.86 | 80 |
| My professors were engaged for my remote course work. | 18 | 22.5% | 20 | 25.0% | 42 | 52.5% | 3 | 2.3 | 0.82 | 80 |
| My remote instruction was worth the amount I paid. | 54 | 69.2% | 12 | 15.4% | 12 | 15.4% | 1 | 1.5 | 0.75 | 78 |
| I was satisfied with my remote course work. | 40 | 50.6% | 17 | 21.5% | 22 | 27.8% | 1 | 1.8 | 0.86 | 79 |

*Mdn*: Median. *M*: Mean. *SD*: Standard Deviation. *n*: sample size.

Results shown in Tables 2 and 3 suggest differences in students' perceptions about in-person and remote learning. Additional details about the remote learning experience were explored through the questionnaire, and Tables 4–6 address these specific aspects of remote instruction. On a positive note, over 80% of respondents in Table 4 agreed or strongly agreed that their internet access and internet speed were adequate for remote instruction. However, five to 10 percent disagreed or strongly disagreed with these statements, which poses concern for a small but noteworthy minority of students that colleges and universities will need to address. Across questionnaire sections, some similar or exact items were asked in order to better establish reliability and validity of students' responses. Again, only half of respondents agreed or strongly agreed that their professors' instruction and assessment translated to remote learning, and close to two-thirds of respondents disagreed that remote instruction was worth the amount they paid ($n = 52$, 63.8%). Student perceptions of their professors' efforts during in-person and remote instruction were interesting. Just over one-third of student respondents agreed or strongly agreed that their professors worked harder to provide remote instruction than in-person instruction, while 57.9% ($n = 44$) agreed or strongly agreed that their professors worked harder to provide in-person instruction than remote instruction. Student perception in these areas may not reflect the reality of professors' efforts to administer remote instruction, yet their perceptions of professor effort are a meaningful take away message.

**Table 4.** Initial Perceptions about Remote Learning.

| Item. | Strongly Disagree/Disagree | | Neutral | | Agree/Strongly Agree | | Summary Statistics | | | |
|---|---|---|---|---|---|---|---|---|---|---|
| | *N* | % | *N* | % | *n* | % | *Mdn* | *M* | *SD* | *n* |
| My internet access was adequate for remote instruction. | 4 | 5.1% | 5 | 6.3% | 70 | 88.6% | 3 | 2.8 | 0.49 | 79 |
| My internet speed was adequate for remote instruction. | 9 | 11.4% | 4 | 5.1% | 66 | 83.5% | 3 | 2.7 | 0.66 | 79 |
| The instructional methods used by my professors translated to remote instruction. | 22 | 27.5% | 17 | 21.3% | 41 | 51.3% | 3 | 2.2 | 0.86 | 80 |
| The assessment methods used by my professors translated to remote instruction. | 18 | 22.5% | 20 | 25.0% | 42 | 52.5% | 3 | 2.3 | 0.82 | 80 |
| My professors worked harder to provide remote instruction than in-person instruction. | 29 | 37.2% | 22 | 28.2% | 27 | 34.6% | 2 | 2.0 | 0.85 | 78 |
| My professors worked harder to provide in-person instruction than remote instruction. | 17 | 22.4% | 15 | 19.7% | 44 | 57.9% | 3 | 2.4 | 0.83 | 76 |
| My remote instruction was worth the amount I paid. | 51 | 63.8% | 17 | 21.3% | 12 | 15.0% | 1 | 1.5 | 0.75 | 80 |
| I was satisfied with my remote instruction. | 36 | 45.0% | 24 | 30.0% | 20 | 25.0% | 2 | 1.8 | 0.82 | 80 |

*Mdn*: Median. *M*: Mean. *SD*: Standard Deviation. *n*: sample size.

Table 5 provides results for items specifically written to address student effort and engagement. Again, results show that the vast majority of students attended remote classes, and while they may not have shown their cameras, close to 80% of respondents *disagreed* that they marked themselves as attending but did NOT participate in their remote course work. A majority (68%) indicated they tried their best and reported their professors tried their best, and close to three-fourths of respondents reported they felt accountable to both themselves ($n = 64$, 77.5%) and their professors ($n = 60$, 75.0%) during remote instruction. Lastly, only half of the students ($n = 40$, 50.6%) agreed to some degree that they actively engaged in their remote classes while a quarter ($n = 20$, 25.3%) disagreed or strongly disagreed with this statement.

**Table 5.** Student Effort and Engagement.

| Item. | Strongly Disagree/Disagree | | Neutral | | Agree/Strongly Agree | | Summary Statistics | | | |
|---|---|---|---|---|---|---|---|---|---|---|
| | *n* | % | *n* | % | *n* | % | *Mdn* | *M* | *SD* | *n* |
| I tried my best during remote instruction. | 14 | 17.5% | 11 | 13.8% | 55 | 68.8% | 3 | 2.5 | 0.78 | 80 |
| My classmates tried their best during remote instruction. | 23 | 31.5% | 16 | 21.9% | 34 | 46.6% | 2 | 2.2 | 0.88 | 73 |
| I attended remote classes. | 2 | 2.5% | | | 77 | 97.5% | 3 | 2.9 | 0.32 | 79 |
| I actively engaged in remote classes. | 20 | 25.3% | 19 | 24.1% | 40 | 50.6% | 3 | 2.3 | 0.84 | 79 |
| I felt accountable to myself to complete remote coursework. | 10 | 12.5% | 8 | 10.0% | 62 | 77.5% | 3 | 2.7 | 0.70 | 80 |
| I felt accountable to my professors to complete remote coursework. | 6 | 7.5% | 14 | 17.5% | 60 | 75.0% | 3 | 2.7 | 0.61 | 80 |
| I marked myself as attended but did NOT participate in my remote course work. | 60 | 77.9% | 6 | 7.8% | 11 | 14.3% | 1 | 1.4 | 0.72 | 77 |
| My professors tried their best during remote instruction. | 11 | 13.9% | 14 | 17.7% | 54 | 68.4% | 3 | 2.5 | 0.73 | 79 |

*Mdn*: Median. *M*: Mean. *SD*: Standard Deviation. *n*: sample size.

Table 6 presents results related to a variety of topics including student needs, behavior, interest in online classes, and perceived effort compared to their instructors. Over 90% of the students also agreed or strongly agreed that they contacted at least one professor independently during remote instruction, but overall, students reported that their needs were not met during remote instruction, and three-fourths disagreed they would take more online-classes. Results associated with student perception of effort were again interesting, with 41.6% agreeing they worked harder than their professors during remote instruction and 25% agreeing that their professors worked harder than them during remote instruction. These mixed responses suggest varied rather than common perceptions among respondents regarding professors' effort. Additional research is needed about the number of hours spent by teachers in planning instruction, teaching, and addressing technical issues; as well as time spent on learning activities by students. This type of additional information would better serve conclusions about student and professor effort and workload.

**Table 6.** Student Needs, Behavior, and Perception of Effort.

| Item. | Strongly Disagree/Disagree | | Neutral | | Agree/Strongly Agree | | Summary Statistics | | | |
|---|---|---|---|---|---|---|---|---|---|---|
| | *n* | % | *n* | % | *n* | % | *Mdn* | *M* | *SD* | *n* |
| My individual needs were met through remote instruction. | 36 | 45.0% | 16 | 20.0% | 28 | 35.0% | 2 | 1.9 | 0.89 | 80 |
| I contacted at least one professor independently during remote instruction. | 5 | 6.3% | 1 | 1.3% | 74 | 92.5% | 3 | 2.9 | 0.50 | 80 |
| I would take more online classes if I lived in my campus housing. | 54 | 74.0% | 6 | 8.2% | 13 | 17.8% | 1 | 1.4 | 0.78 | 73 |
| I worked harder than my professors during remote instruction. | 19 | 24.7% | 26 | 33.8% | 32 | 41.6% | 2 | 2.2 | 0.80 | 77 |
| My professors worked harder than me during remote instruction. | 26 | 35.1% | 29 | 39.2% | 19 | 25.7% | 2 | 1.9 | 0.78 | 74 |

*Mdn*: Median. *M*: Mean. *SD*: Standard Deviation. *n*: sample size.

Table 7 presents results related to student ethics. Results related to ethical behavior during remote instruction pose some concerns, as approximately one-third or more of respondents agreed that they collaborated with classmates or individuals not in their classes to complete individually assigned remote course work and tests, and cheated based on their professors' definitions of cheating ($n = 33$, 42.9%) and their personal definition of cheating ($n = 26$, 33.3%). Differences in these percentages suggest individuals may have different definitions than their professors of cheating. Nonetheless, these results related to ethical behavior during remote instruction are of concern.

**Table 7.** Student Ethics.

| Item. | Strongly Disagree/Disagree | | Neutral | | Agree/Strongly Agree | | Summary Statistics | | | |
|---|---|---|---|---|---|---|---|---|---|---|
| | *n* | % | *n* | % | *n* | % | *Mdn* | *M* | *SD* | *n* |
| I collaborated with classmates to complete individually assigned remote course work. | 29 | 36.7% | 5 | 6.3% | 45 | 57.0% | 3 | 2.2 | 0.95 | 79 |
| I collaborated with people not in my classes to complete individually assigned remote course work. | 53 | 67.1% | 3 | 3.8% | 23 | 29.1% | 1 | 1.6 | 0.91 | 79 |
| I collaborated with classmates to complete tests in remote classes. | 48 | 63.2% | 5 | 6.6% | 23 | 30.3% | 1 | 1.7 | 0.91 | 76 |
| Per my professors' definitions of cheating, I cheated at least once in remote classes. | 40 | 51.9% | 4 | 5.2% | 33 | 42.9% | 1 | 1.9 | 0.98 | 77 |
| Per my definition of cheating, I cheated at least once in remote classes. | 44 | 56.4% | 8 | 10.3% | 26 | 33.3% | 1 | 1.8 | 0.92 | 78 |

*Mdn*: Median. *M*: Mean. *SD*: Standard Deviation. *n*: sample size.

### 3.2. Questionnaire Results by Demographic Subgroups

The researchers investigated differences in questionnaire results for several demographic variables collected in the current study using Chi-square tests of independence at alpha = 0.01. Table 8 summarizes results of analyses of the demographics including student level (undergraduate versus graduate), type of institution (public versus private), student status (commuter versus non-commuter), and size of institution (less than 5000, 5000-less than 20,000, and more than 20,000). In order to satisfy the assumptions of the Chi-Square tests given the sample size in the current study, the categories of agree and strongly agreed were combined; as were categories of disagree, strongly disagree, and neutral, so the Chi-Square tests were based on two response categories. Additionally, all demographic variables had two categories with the exception of size of institution, which had three response categories, accounting for the degrees of freedom listed in Table 8.

Generally speaking, results did not show systematic differences in the response percentages based on any of the demographic subgroups, as only a total of 14 significant differences across items and all demographic variables were found. No differences were found on items designed to measure *General Perceptions of In-person Learning* and *Student Effort and Engagement*. Additionally, no differences were found in responses across the demographics of type of institution (public versus private) or size of school institution (less than 5000, 5000-less than 20,000, and more than 20,000). Table 7 shows the item subscale, demographic variable, Chi-square test statistic, degrees of freedom and p-value for all items that resulted in different percentages of agreement based on any of the demographics.

Several differences were found in reported level of agreement based on student level of undergraduate or graduate, as shown in Table 8, and two differences were found based on student status (commuter versus non-commuter). With so few differences being found between commuters and non-commuters, it is possible that the differences that were found were actually confounded by the fact that a large number of the commuters were graduate students, rather than being strictly due to students' status as commuters. When

differences were found among demographic subgroups, more undergraduates disagreed or remained neutral they were challenged and engaged in coursework, that their individual needs were met, and that their instruction was worth the amount they paid; and more undergraduates agreed that they collaborated and cheated more than graduate students. Both undergraduate commuters and graduate students may not have established a cohort with whom they could work together as did the undergraduate students who had been living on campus, accounting for differences in items related to ethical behavior. Additional research to explore the perceptions and behaviors of various subgroups of students could shed more light on student behavior; however, our results generally suggested little differences in item response based on demographics for the vast majority of items on the questionnaire.

**Table 8.** Differences in Percentages across Student Level (Undergraduate vs. Graduate).

| Item Subscale. | Demographic | Item | $\chi^2$ | *df* | *p* |
|---|---|---|---|---|---|
| General Perceptions of Learning (Remote Learning) | Student Level UG versus G | I was challenged by my remote course work. | 17.88 | 1 | <0.001 |
| | | I was engaged in my remote courses. | 10.82 | 1 | 0.001 |
| | | I cheated during my remote courses | 13.97 | 1 | <0.001 |
| | | My remote instruction was worth the amount I paid | 17.77 | 1 | <0.001 |
| | | I was satisfied with my remote course work | 8.29 | 1 | 0.004 |
| | Student Status Commuter versus Noncommuter | I was challenged by my remote course work | 6.59 | 1 | 0.01 |
| | | My remote instruction was worth the amount I paid. | 10.21 | 1 | 0.001 |
| Initial Perceptions about Remote Learning | Student Level UG versus G | My professors worked harder to provide in-person instruction than remote instruction. | 8.33 | 1 | 0.004 |
| | | My remote instruction was worth the amount I paid. | 8.42 | 1 | 0.004 |
| | | I was satisfied with my remote instruction. | 13.80 | 1 | <0.001 |
| Student Needs | Student Level UG versus G | My individual needs were met through remote instruction | 11.76 | 1 | <0.001 |
| Student Ethics | Student Level UG versus G | I collaborated with classmates to complete tests in remote classes. | 7.02 | 1 | 0.008 |
| | | Per my professors' definitions of cheating, I cheated at least once in remote classes. | 16.31 | 1 | >0.001 |
| | | Per my definition of cheating, I cheated at least once in remote classes. | 10.86 | 1 | >0.001 |

$\chi^2$ Chi Square; df: degrees of freedom; *p*: *p*-value.

### 3.3. Open-Ended Responses

Students were asked three open-ended items on the questionnaire. First, students were asked to describe their experience with remote instruction during the Spring 2020 semester. Sixty-two of the 80 respondents provided a response. Thirty-one of these were negative comments, 24 neutral or mixed, and only seven were positive in nature. Negative comments related to feeling disengaged; course content not being conducive to online learning; students not responding well to online learning; lack of planning by

professors; and general disorganization. Positive comments related to professors working hard; students being able to continue their degree coursework during a pandemic; increased engagement; or that they just liked the experience. Mixed responses pointed out both these positive and negative aspects.

Second, students were asked what they liked about their experience with remote instruction during the Spring 2020 semester and 64 provided a response. The most common responses related to the flexible schedule or self-paced nature of the instruction ($n = 17$); enjoyment associated with the at-home setting that was more relaxed and did not require a commute ($n = 15$); or the flexibility or engagement of some of their professors ($n = 8$). Three individuals provided a positive response, which included collaborating with technology, attending live classes, and experiencing success in their classes. Singular respondents liked aspects such as online tests, that they surprised themselves with their focus, and that attending helped their mental status or flippant responses such as they liked that they did better due to cheating. Eight individuals reported they liked "nothing" about remote instruction.

Last, students were asked to provide the suggestions they have for their professors that could improve your experience with remote instruction based on the Spring 2020 semester, and 54 respondents commented. Common responses include wanting professors to use more varied or better instructional strategies ($n = 21$); be more understanding ($n = 9$); include less busy work ($n = 6$); mix live and asynchronous classes ($n = 4$); and have clearer assignments and course shells ($n = 3$).

### 3.4. Interview Themes

Semi-structured interviews were conducted with a purposeful sample of ten undergraduate students enrolled full-time during the Spring 2020 semester. Inclusionary criteria included undergraduate students who experienced a rapid transition to virtual, online learning. The interview protocol, created by the researchers, sought to examine issues related to impressions of remote learning more in depth for undergraduate students who were forced to switch abruptly from in-person to remote learning. Six themes were evident from the qualitative data which included; *juggling act, overwhelming sense of frustration, lack of engagement, motivation and accountability, we want our voices heard, cheating to finish*, and *professor qualities*.

#### 3.4.1. Theme 1: Juggling Act

Theme one, *Juggling Act*, was apparent from the interviews. All participants discussed transitioning from on campus, face-to-face instruction to virtual instruction was challenging. In particular, six participants mentioned the difficulty of juggling learning from home while family members were also working from home. P1 stated, "I really thought I would make this work to my advantage, but I was really trying to juggle taking classes online while my parents were also in the house working. Then, my brother came home to work." Likewise, P4 mentioned the difficulty with learning at home with a full house. She stated, "We had three people trying to work at home at the same time. It was difficult for all of us, I think." Five participants also mentioned distractions as an obstacle to learning at home. Students' struggles to transition to online, remote learning from a traditional on-ground, in-person pedagogical approach were documented in the interviews. Importantly, all students reported that outside influences impacted their ability to concentrate and learn in a new environment.

#### 3.4.2. Theme 2: Overwhelming Sense of Frustration

The second theme, *Overwhelming Sense of Frustration*, emerged from all ten participants. P2 stated, "I was so overwhelmed because I had so much work and I have never taken a class online so I didn't know what to expect. I was scared I wouldn't pass my classes." The sense of frustration was apparent in P6's response. "I don't know what was more frustrating? Me trying to learn or some of my professors trying to teach." P7 shared, "at first

it seemed cool, but I quickly realized it was not." The difficulty of working remotely, and often in other time zones, was also a struggle for participants. P9 discussed her difficulties with living in a time zone different from her university. She added, "I had to log in for an 8 a.m. class and it was 5 a.m. on the west coast." Likewise, P7 and P10 also mentioned classmates who were in international time zones making attending virtual class in real time difficult if not impossible. These additional pressures added stress and frustration to the already arduous situation.

### 3.4.3. Theme 3: Lack of Engagement, Motivation and Accountability

All interview participants mentioned their *Lack of Engagement, Motivation, and Accountability* when they left campus to learn at home. P1 stated, "My level of engagement depended on the class, but I was less engaged at home than I would be at school. I also was worried about me and my family getting sick so that was distracting." Likewise, P2 stated, "I always volunteer in class, but online I played games on my phone or caught up on my Snap Chat. Some professors were better than others at making us accountable and engaged." Most (*n* = 9) participants also discussed their lack of motivation when classes moved to remote learning. "Once my trip to Serbia was canceled and my sorority events were canceled, I lost motivation to even do anything. I laid in bed and did all my assignments. It is sad now that I think about it," said P4. Moreover, P6 stated, "When you are in person, you are motivated to take notes and participate. By week 15, I was just getting by to finish the semester." P8 commented on their lack of motivation, "I had too many distractions. I mean I could sit on my phone and watch Tik Tok." P9 added, "I was on my computer for hours, and messages would be coming up from friends. It was easy to get distracted. Plus I was at home, just not in a school mindset." Clearly, the participants' lack of engagement with course content and instructors, motivation to succeed and participate were adversely affected. Few participants' courses required accountability for course content or active engagement in class activities; thus, motivation was lacking.

### 3.4.4. Theme 4: We Want Our Voices Heard

Eight of the ten participants mentioned *Wanting to be Heard* in some capacity. For instance, P2 stated," We just wanted professors to keep it as normal as possible for us. It all happened so fast and we were scared. We wanted professors to understand where we were coming from." P5 also stated, "We wanted professors to be patient with students. You only know us as students. We were sent home to different environments with personal home lives that aren't perfect, and we may not feel comfortable reaching out." P8 offered advice for professors, "Be as understanding of us as we are understanding of them." P10 added, "Our college had no online classes prior to the move in March so I was very nervous, but then it felt like we're all in this together so that was helpful." All participants mentioned the phrase "being heard" was important collectively. Individual differences, in participants' opinions, were not taken into account by instructors.

### 3.4.5. Theme 5: Cheating to Finish

Seven of the ten participants admitted to cheating so they could "just finish" the semester, supporting the *Cheating to Finish* theme. P2 commented, "Some professors just gave busy work so we would all get on FaceTime and cheat just to earn the points." This sentiment was echoed by others. P5 stated, "Friends helped me with my tests and quizzes a lot. We were all frustrated with having to go home, and we just wanted to be done and get the grade." P8 added, "I could look stuff up on the internet, and I collaborated with others in class. I know maybe technically it was cheating, but I don't always understand the first time and it was nice to be able to talk to others who helped me to understand things." Nine of the ten participants mentioned the concept of collaboration as a means to an end to consummate the semester.

### 3.4.6. Theme 6: Professor Qualities

Seven of the ten interview participants observed the transition, either negative or positive, was dependent on the *Professor Qualities*. They discussed how the variance was great. P7 said, "I mean some of these professors are dinosaurs. If they can't understand, how can we? I had one who was very nice, but kept sending assignments in pdf form and it would take lots of emails back and forth explaining to her that we could not write on them, and then assignments were late." P9 commented that even if professors were not tech savvy, they had to be willing to learn. She continued that loading PowerPoint presentations on a learning management system was unhelpful. All participants stated that they missed the in-person engagement that occurs in a face-to-face, on ground classroom. They all shared that they felt their learning suffered by not attending in-person instruction, and missing out on others sharing ideas, with professors guiding discussion.

## 4. Discussion

Questionnaire data provide some insight about students' perceptions before and after their abrupt change from in-person to remote learning. Students were initially satisfied and engaged in course work; and results from both questionnaire and interview data suggest students' perceptions about satisfaction and engagement both decreased with their change to remote learning. Open-ended responses provided additional insight about these areas, and data suggested that personal obstacles and barriers to success took precedence over academic accomplishments.

Furthermore, interview data allowed topics to be explored more in-depth, and students' concerns and frustrations became apparent. With both faculty and students facing abrupt changes with little time for planning, students were nervous about the switch, as they had not previously engaged in online or remote learning. Several students reported receiving low quality remote instruction, including emailed assignments; but even for live remote classes, students were distracted by other technology or home experiences, and were less engaged. To make up for their lack of focus and engagement, they seemed to collaborate with others on assignments including tests and quizzes, and justify their responses with their feelings of being overwhelmed and wanting to get through the semester. Cheating was often viewed as a means to an end.

Additionally, instructors seemed to communicate less with students and hold them less accountable. Data related to student accountability differed from the questionnaire data and interview data. While approximately 75% of survey respondents indicated they felt accountable to themselves and their instructors, interview data suggested students felt less accountability during remote learning.

Results from questionnaire data in the current study suggest that perceptions of students who began the Spring 2020 semester engaged and satisfied with their instruction became less positive with the switch to remote instruction, which may be a pedagogical challenge for some instructors (Ali 2020). Moreover, students' self-regulating processes of self-motivation and self-directed learning can be challenging (Abdullah and Ward 2016; Fredricks et al. 2004; Tichavsky Lisa P. et al. 2015). Students' perceptions, who were originally attending classes in-person and abruptly switched to virtual instruction, became more negative regardless of the level of the student, type of institution, student status as commuter versus non commuter, or size of institution. Students reported attending classes but fewer reported being engaged and challenged, and more, especially undergraduate students, reported cheating during remote instruction. Student perceptions about how hard instructors worked during remote instruction were mixed.

Data from in-depth interviews suggested students were aware they were less engaged, less motivated, and lacked accountability; and they knew they cheated on classwork. The abrupt switch to remote instruction took a toll on students, as they experienced overwhelming frustration with the switch (Hodges et al. 2020). They also reported needing more flexibility from instructors, and missing the engagement they normally feel in their in-person classes (Richardson and Newby 2010).

These results seem to align with existing literature about distance matters in delivering course work (Billings et al. 2001; Olson and Olson 2000; Wells and Dellinger 2011). For students, who were already feeling isolated and frustrated, remote courses seemed to lack connectedness, leaving them with additional feelings of isolation and frustration. Additionally, it seems there was a lack of common ground as students and faculty alike struggled to make sense of the new learning environment. Furthermore, a mutual readiness for collaboration was non-existent as neither group, students or faculty had a choice in moving to virtual course delivery. Lastly, there was little to no time to address technology needs and course work was being redefined in real time, leaving everyone a bit adrift. Wells and Dellinger (2011) have suggested that the quality of instruction can be more of an influence on learning that the type of course delivery. However, given that faculty were generally given a window of days to weeks to convert course material and delivery, it stands to reason that the majority of faculty were struggling along with their students.

Much of the current literature related to online learning addresses students who intentionally choose on-line programs. However, the students of 2020, including those in the current study, were all initially participating in live, in-person classes, and were abruptly forced into remote learning situations. Therefore, our results reflect a different population than is addressed in much of the existing literature. However, it is possible to offer new ideas about students' experiences and needs. Additionally, as more universities worldwide will be utilizing remote and online learning, the need to understand strengths and weaknesses is critical (Chan 2020). More students than ever will likely be learning online and faculties must be more than just "tech-savvy." Faculties will need to be prepared to engage students and enhance their online learning experiences from a pedagogical standpoint (Chan 2020). As such, our research results in several key findings:

1.  Students engaged in remote instruction, who have not chosen this type of learning modality, will require more explanations of expectations and more accountability.
2.  Students miss engagement with their professors and peers.
3.  Instructors need to be aware that students may collaborate with others in remote learning situations more than they would during in-person instruction and choose assignments and grading methods in light of these behaviors.
4.  Access to reliable technology is an issue for a small but relevant percentage of students at the university level.

Additionally, we offer recommendations based on our results:

1.  Students who are used to in-person instruction need guidance on how to engage effectively in remote and on-line learning.
2.  Instructors must develop and implement remote learning activities to engage students to the same degree as in-person activities and discussions.
3.  Flexibility on the part of students and instructors is needed, yet should not override accountability or expectations of ethical behavior for either group.

Limitations of the study include sample size and the mainly singular gender participation. While the current sample is limited in size and representativeness is unknown, surveying more systematically now that the college students have had remote or hybrid classes for two terms and faculty had time to learn and plan will show the extent to which our baseline data hold true. Due to the nature of the timeline of the study, snowball sampling was implemented for interview participants, rather than strict criterion-based sampling.

The survey sample was relatively small and while several personal and university-level demographic variables were represented, the sample may not be representative of all students. The topic was timely and required some haste, at a time when most universities were struggling with keeping students safe, moving to remote instruction, and dealing with a variety of other unforeseen issues. Given these constraints, the researchers decided recruiting directly from universities would not result in a higher yield. Additionally, the pandemic made face-to-face recruitment an impossibility. Therefore, the generalizabil-

ity of results is unknown. Additionally, the interview sample consisted of all female, undergraduate students who were willing to participate.

Recommendations for future research include examining both genders as differences may exist as the sample was not representative. Investigating the transition to virtual learning across the course of the pandemic would be valuable as many universities are still utilizing a hybrid format for many courses. Moreover, the examination of students' attitudes and perceptions may be different from the initial transition in March of 2020. Additionally, students in this study perceived professors as "working harder" in face-to-face, on-ground classes. Examining the amount of time professor spend on structuring learning activities, teaching, dealing with technology issues, and meeting with students outside of class in a remote versus face-to-face courses might lead to interesting discussions of professional work load. Furthermore, what does "working harder" mean to college students? Perhaps they perceive faculty engagement and connection as a primary aspect of "hard work." Given it is an area documented to be more difficult to achieve with online and remote course work, it would be interesting to understand how this plays into student perceptions of professor effort. Finally, exploring faculty comfort level with technology as Institutes of Higher Education move into the year 2021 might provide administrators input and insight for professional development as it seems remote learning is here to stay.

In light of the recent COVID-19 pandemic, it is imperative that universities and their faculty members learn to be flexible, intentional, and consider means to engage and motivate students in the virtual environment. While student engagement neither guarantees accountability nor decreases student-cheating behaviors, it could indeed help students retain content. Moreover, examination of instructional strategies used by faculty while teaching virtually could help students succeed in such an environment. It is imperative that universities, and the faculty, continue to examine the ways in which this pandemic affected both undergraduate and graduate students.

**Author Contributions:** Data curation, S.W.P., M.A.H. and C.B.; Formal analysis, M.A.H.; Investigation, S.W.P. and C.B.; Methodology, M.A.H. and C.B.; Writing—original draft, S.W.P.; Writing—review & editing, S.W.P. and C.B. All authors have read and agreed to the published version of the manuscript.

**Funding:** This research received no external funding.

**Institutional Review Board Statement:** Robert Morris University's Institutional Review Board approved the study on 14 May 2020 (IRB#202004162364).

**Informed Consent Statement:** Informed consent was obtained from all subjects involved in the study.

**Data Availability Statement:** Data is housed with the researchers.

**Conflicts of Interest:** The authors declare no conflict of interest.

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
