# Peer review of "COVID-19 Campus Closures in the United States: American Student Perceptions of Forced Transition to Remote Learning"

_socsci, doi:10.3390/socsci10020062_

Round 1

Reviewer 1 Report

1. The title is correct. It focuses on the centres of interest of research.
2. In the summary, the objective is defined. However, the methodology employed, it is reported that questionnaires and semi-structured interviews were used, but the methodology must inform about the name, brand or type of tool used, as well as the software and the survey that has been carried out should be highlighted; or, on the contrary, if the questionnaire has been elaborated by the researchers. On the other hand, the abstract should inform whether it is a qualitative or quantitative study. As for the results, the most relevant ones are reported. On the other hand, it is not reported where the study can be useful, it would be convenient to give some information about it. I also miss some comments on the originality of the study. It would also be appropriate to remove bibliographic references from the abstract.
3. Introduction is OK, but the rest of the structure of the headings should be changed. It is logical to continue with the methodological section and then with the results and discussion.
4. The rest of the structure of the headings should be changed. It is logical to continue with the methodological section and then with the results and discussion. The authors are asked to follow this structure.
5. The methodology is appreciated in a correct way. It presents the method used for the study, as well as the conditions of the research.
6. As for the results, the study's finding is described objectively and in a clear and precise manner.
7. However, it would be more correct to have added in the discussion block a combination of its findings in relation to those identified in the literature review, and to place them within the context of the theoretical framework that supports the study. I can only find one sentence which is "These results align with much of the existing literature, and the same time, offer new ideas about students' experiences and needs", which is very short and concise without support in the theoretical framework of the study.
8. The block of limitations included in the manuscript is appreciated. But it is necessary that they make some recommendations on the need for future research taking into account the limitations of the study..

Author Response

Reviewer 1

Methodology not mentioned in summary/abstract

Added “QUAN-QUAL descriptive mixed methods study” to the abstract.

Statistical software not mentioned in abstract

Added “Results, analyzed using SPSS (QUAN) and inter-coder agreement (QUAL)”

Review bibliographic references from abstract

Completed-deleted citation

Suggestions for corrections in attached document for English syntax, punctuation, and typos

Cannot find the attached document

Introduction is OK, but the rest of the structure of the headings should be changed. It is logical to continue with the methodological section and then with the results and discussion.

Moved methods per the reviewer suggestion (the journal template placed methods in that section). All table numbers changed to reflect this change.

The rest of the structure of the headings should be changed. It is logical to continue with the methodological section and then with the results and discussion. The authors are asked to follow this structure.

All headings were changed to reflect moving the methods section forward.

However, it would be more correct to have added in the discussion block a combination of its findings in relation to those identified in the literature review, and to place them within the context of the theoretical framework that supports the study. I can only find one sentence which is "These results align with much of the existing literature, and the same time, offer new ideas about students' experiences and needs", which is very short and concise without support in the theoretical framework of the study.

Completed

The block of limitations included in the manuscript is appreciated. But it is necessary that they make some recommendations on the need for future research taking into account the limitations of the study.

Added recommendations for future research based on the limitations of the study

Reviewer 2 Report

Abstract is well written with a minor comma error: in line 19, no comma is needed after "instruction." 

Thank you for the opportunity to review this article.  It is very timely and interesting.  It is well written, as well.  The main issue I see is the Materials and Methods section should come before the results section.  I feel it would make the article flow much better. 

Thesis, aims, purpose of article

Line 26: “globe” instead of “glove”

Listing the specific research questions would be helpful to the reader.

Line 61: Instead of “the review,” perhaps it would be easier for the reader to understand if it said, “A review of the literature.”

Technology

This section seems to need to begin with a statement transitioning from the last paragraph.

Results

It seems that a methodology section is needed.  *I found it at the end of the article.  It would be helpful to move the before the results section.

Has the questionnaire been used before?  If so, what permission was obtained?  If not, were the

questions piloted? 

Interview Themes

Line 525: a comma instead of a period is needed after “students,” as it is not a complete sentence.  As such, “There” on the next line does not need a capital. 

I understand that this is research on remote learning during the pandemic, but is there any other research on online learning that supports the findings?  If so, it would likely be beneficial to include that information in the results/discussion.

Author Response

Reviewer 2

Abstract is well written with a minor comma error: in line 19, no comma is needed after "instruction." 

Completed: I think the reviewer was referring to the word “initially”

he main issue I see is the Materials and Methods section should come before the results section.  I feel it would make the article flow much better. 

Moved methods per the reviewer suggestion (the journal template placed methods in that section). All table numbers changed to reflect this change.

Line 26: “globe” instead of “glove”

Completed

Listing the specific research questions would be helpful to the reader.

Research questions have been added (lines 73-76)

Line 61: Instead of “the review,” perhaps it would be easier for the reader to understand if it said, “A review of the literature.”

Completed

This section seems to need to begin with a statement transitioning from the last paragraph.

Added, “Current discussion seems to be generally centered on technology, faculty perceptions, and student perceptions (including student mental health).”

It seems that a methodology section is needed.  *I found it at the end of the article.  It would be helpful to move the before the results section.

Moved methods per the reviewer suggestion (the journal template placed methods in that section). All table numbers changed to reflect this change.

Has the questionnaire been used before?  If so, what permission was obtained?  If not, were the questions piloted? 

This questionnaire was researcher-developed and was not piloted.

Line 525: a comma instead of a period is needed after “students,” as it is not a complete sentence.  As such, “There” on the next line does not need a capital. 

Can’t find

 I also could not find.

 I searched ”there” and could not find

I understand that this is research on remote learning during the pandemic, but is there any other research on online learning that supports the findings?  If so, it would likely be beneficial to include that information in the results/discussion.

Completed, Included in Intro. and Discussion

Reviewer 3 Report

This paper has merit and is worthy of publication pending minor revisions. The authors have conducted a much needed study in a thorough manner using a sound mix-method methodology (which should appear under section 2) and presenting a detailed reporting of the results. The placement of this research within the existing literature could be made clearer in the Introduction section by emphasising the scarcity of research in this area (understandably for good reasons) and therefore the originality of the present study. Similarly, the discussion section (which should be deferred to section 3 and not 2!) could be made stronger by emphasising how other studies can build on these results and further refine them.

I am attaching the original paper with suggestions for corrections (English syntax, punctuation, typos) and clarifications when needed.

With best wishes.

Author Response

Reviewer 3

The authors have conducted a much needed study in a thorough manner using a sound mix-method methodology (which should appear under section 2)

Completed

the placement of this research within the existing literature could be made clearer in the Introduction section by emphasising the scarcity of research in this area (understandably for good reasons) and therefore the originality of the present study. 

Completed

the discussion section (which should be deferred to section 3 and not 2!) could be made stronger by emphasising how other studies can build on these results and further refine them.

Completed

All grammatical/typos fixed

Completed from track changes provided